# The Genetics of Thoracic Aortic Aneurysms and Dissection: A Clinical Perspective

**DOI:** 10.3390/biom10020182

**Published:** 2020-01-24

**Authors:** Nicolai P. Ostberg, Mohammad A. Zafar, Bulat A. Ziganshin, John A. Elefteriades

**Affiliations:** 1Aortic Institute at Yale-New Haven Hospital, Yale University School of Medicine, New Haven, CT 06510, USA; nostberg@stanford.edu (N.P.O.); mohammad.zafar@yale.edu (M.A.Z.); bulat.ziganshin@yale.edu (B.A.Z.); 2Department of Cardiovascular and Endovascular Surgery, Kazan State Medical University, 420012 Kazan, Russia

**Keywords:** thoracic aortic aneurysm and dissection (TAAD), genetics, genetic screening, syndromic TAAD, non-syndromic TAAD

## Abstract

Thoracic aortic aneurysm and dissection (TAAD) affects many patients globally and has high mortality rates if undetected. Once thought to be solely a degenerative disease that afflicted the aorta due to high pressure and biomechanical stress, extensive investigation of the heritability and natural history of TAAD has shown a clear genetic basis for the disease. Here, we review both the cellular mechanisms and clinical manifestations of syndromic and non-syndromic TAAD. We particularly focus on genes that have been linked to dissection at diameters <5.0 cm, the current lower bound for surgical intervention. Genetic screening tests to identify patients with TAAD associated mutations that place them at high risk for dissection are also discussed.

## 1. Introduction

Thoracic aortic aneurysm (TAA) is a dangerous, deadly, and silent disease that is notoriously difficult to detect and diagnose prior to complications [1]. It is believed that only 5% of TAA are symptomatic prior to dissection or rupture and even when dissection presents with symptoms, less than 50% are promptly diagnosed in the emergency department prior to death [2]. Together, TAA and abdominal aortic aneurysms (AAA) constitute the 17th leading cause of death in patients over the age of 65 [3]. Furthermore, many TAA dissections tend to be misdiagnosed as myocardial infarctions, suggesting a higher true mortality rate [4]. Yet, if TAAs are diagnosed and managed surgically prior to dissection, survival nearly matches population controls, with limited complications [5,6]. Therefore, early diagnosis and management remain a critical component for limiting mortality from TAA. 

Due to the difficulty of diagnosing TAA based on symptoms alone, efforts have been undertaken to determine which patients are at higher risk of harboring TAA and therefore, warrant preemptive screening for early diagnosis of aneurysm, monitoring, and appropriately tuned surgical intervention [7]. Because more than 20% of patients report a positive family history, the genetics of thoracic aortic aneurysm and dissection (TAAD) has been extensively investigated as a potential avenue for both diagnosis and risk stratification [8]. 

Traditionally, TAAD is divided into syndromic—with other organ system abnormalities other than the aorta—and non-syndromic—with no other systemic abnormalities present. Non-syndromic disease is then further subdivided to include familial non-syndromic TAAD, where one or more family members are diagnosed with TAAD, and non-familial TAAD, where no other family members are affected [9]. However, in both syndromic and non-syndromic cases, often, only a single gene is mutated and that same mutated gene can produce both syndromic and non-syndromic TAAD [10]. Moreover, there is significant variation in the severity of TAAD presentation, even within the same family [11]. Therefore, this classification system provides a convenient hierarchy for discussing the genetic causes of TAAD, although the true clinical picture is much more complex.

Several causes of syndromic TAAD are well described, such as Marfan syndrome (MFS), Loeys–Dietz syndrome (LDS), and Ehlers-Danlos Syndrome (EDS) [12]. While the existence of systemic features may make diagnosis easier, the wide spectrum of physical manifestations that these diseases can assume makes the task of prompt diagnosis more difficult. A high index of suspicion must be maintained to detect the more mild forms of syndromic manifestations or presentations that overlap with other conditions [13,14]. The prognosis of syndromic TAA cases is generally worse than non-syndromic cases, resulting in modified surgical intervention criteria at aortic diameters less than the typical boundary of a 5.0–5.5 cm aortic diameter [15]. Regardless, syndromic TAAD accounts for only approximately one-fifth of all TAAD cases, meaning that other genes and mechanisms must be implicated in the development of TAAD [2].

Cases of non-syndromic TAA are more prevalent, but identifying these patients can be challenging. Some cases can be explained by mutations in genes involved in syndromic TAAD [16]. Approximately 20% of non-syndromic patients have one affected family member, known as familial TAAD, providing further evidence for a genetic link [17,18,19]. However, this leaves a substantial majority of patients with no cause of syndromic TAAD and no family history—what can be thought of as “random” or sporadic TAAD. More investigation is being undertaken to understand these seemingly random cases of TAAD and some interesting associations have been uncovered. For example, mutations in the causative gene in MFS have also been linked to sporadic TAAD [20]. However, further work needs to be done to fully understand the genetic interactions that lead to both familial and sporadic TAAD.

In this review, we describe the genetic and molecular mechanisms of common syndromic and non-syndromic manifestations of TAAD, examine the clinical management of both classes of TAAD, and discuss the future applications of genetics to guide the screening, diagnosis, and management of patients. Many genes play a role in the development of TAAD and each requires unique considerations and gene specific surgical intervention guidelines (Figure 1). However, unlike multifactorial diseases like atherosclerosis, TAAD tends to be caused by a single base change in a single gene [7].

## 2. Causes of Syndromic TAAD

Syndromic conditions that cause an increased risk of TAAD have been well described for decades, although the causative mutations have only just recently been verified. Typically, syndromic TAAD is linked to dysfunction of the extracellular matrix (ECM) [21], medial smooth muscle cells (SMC) [22], or TGF- β signaling [23]. While several other pathologies are often simultaneously present in these syndromic cases, the most deadly concern typically is vascular, as the weakening of the wall of the aorta can cause dilation, dissection, and potential rupture of the aorta. The three most common and well described diseases linked to syndromic TAAD and their respective pathophysiology are reviewed. Other syndromic causes are briefly discussed.

### 2.1. Marfan Syndrome (MFS)

Microfibrils, which are largely composed of two large glycoproteins, fibrilin-1 and fibrilin-2, constitute a structurally significant component for the integrity of the extracellular matrix throughout the body [24]. In association with essential elastin, microfibrils contribute significantly to the stability and elasticity of the aorta [25]. Mutations in the gene for fibrillin-1, FBN1, have been causatively linked to MFS, an autosomal dominant connective tissue disorder with numerous associated pathologies, including skeletal, ocular, and, of note for this review, cardiovascular abnormalities [26]. Missense mutations in FBN1 were first linked to MFS in 1991 [27]. Knowledge has since grown to include 1847 highly penetrant unique mutations, and it is likely that many more exist (http://www.umd.be/FBN1/—last updated 28 August 2014). Most mutations in FBN1 are heterozygous but recent evidence suggests that both homozygous and compound heterozygous mutations exists in approximately 0.5% of MFS patients [28]. It is important to note that FBN1 mutations have been shown to cause both syndromic (MFS) and non-syndromic TAAD; interestingly, variants in FBN1 were the most frequently mutated gene in a large whole exome sequencing study of syndromic and non-syndromic TAAD patients (Figure 2) [29].

Investigation into these numerous mutations has revealed some genotype–phenotype associations. However, there is still a wide range of phenotypic variability in patients with the same FBN1 mutation, suggesting that complex gene interactions are at play [31]. Distinctions have been drawn between haploinsufficient (HI) mutations and dominant negative (DN) mutations. HI mutations of FBN1 result in only one functional copy of the gene, often the result of frameshift, nonsense, or splice-site mutation. The fibrillin levels decrease, resulting in reduced structural integrity of microfibrils [32]. Dominant negative (DN) mutations that result in a mutated protein lead to large phenotypic variability, depending on the region of the protein that is affected [33]. Evidence suggests that neither mutation class fully captures all of the phenotypic variation seen in FBN1 mutations; both HI and DN mutations cause MFS. However, patients with HI mutations have been shown to have significantly higher rates of aortic dissection, although overall cardiovascular mortality is comparable between the two mutation classes [34]. In addition, DN mutations in functionally significant cystine residues have been linked to higher probability of aortic dilation and dissection [35,36] and those affecting calcium binding exert structural effects on fibrillin assembly [37]. Interestingly, mutations in FBN1 have also been found in about 4% of TAAD patients without typical clinical manifestations of MFS [38]. Recent work has shown that FBN1 hypomorphic mice also displayed similar phenotypes, with increased elastin fragmentation, reduced wall strain, and larger aortic diameters [39].

Fibrilin-1 has also been shown to interact with TGF-β (transforming growth factor-β). TGF-β is a peptide that associates with latency-associated protein (LAP) to form the small latent complex, which, in turn, associated with latency-associated TGF-β binding protein to form the large latent complex (LLC). The LLC targets TGF-β to fibrillin microfibrils, sequestering it within the walls of the aorta [40]. It has been suggested that FBN1 mutations seen in MFS result in impaired TGF-β sequestering [41]. Because increased TGF-β signalling has been linked to TAAD [42], fibrillin-1′s inability to adequately regulate TGF-β may also be a causative or contributing factor to TAAD in MFS patients.

Clinically, patients presenting with MFS are at high risk for developing aortic dissections due to the structural instability of their aortas. MFS patient with FBN1 mutations are also more likely to develop dissections at the aortic root [43]. As with patients without MFS, the rate of dissection tends to correlate with diameter. The annual risk of dissection is 13% in MFS patients with aortic diameters greater than 5.0 cm [44]. The growth rate of these aneurysms has been measured at 0.26 cm/year, and rate of growth increases at larger diameters [44]. Currently, guidelines suggest operation when the aneurysm reaches 4.5–5.0 cm in diameter in MFS patients. Currently, clinical management of MFS patients does not typically vary based on the specific mutation present, due to the aforementioned high degree of phenotypic and behavioral variability across the same genetic mutation. However, certain groups of patients may benefit from surgery at smaller diameters: patients with rapidly growing aneurysms or patients who plan on becoming pregnant are suggested to receive prophylactic aortic repair surgery [45].

In the future, it is likely that surgical treatment for MFS patients will be predicated on the specific FBN1 mutation present, achieving an even higher level of genetically based precision medicine.

### 2.2. Loeys–Dietz Syndrome (LDS)

First described in 2005, Loeys–Dietz syndrome (LDS) is an autosomal dominant disorder characterized by severly increased risk of aortic aneurysms and dissection along with other defects [46]. Although the disease was first thought to be linked to mutations in the transforming growth factor β receptor I (TGFBR1) and transforming growth factor β receptor II (TGFBR2), other mutations have also been linked to LDS, including decapentaplegic homolog 3 (SMAD3) and transforming growth factor β II (TGFB2) [46,47,48]. Due to phenotypic variation across these different mutations and the high risk of aneurysm even without other associated systemic features of LDS, four different subgroups of LDS (LDS1-LDS4) were developed that describe different pathophysiologies and prognoses [49]. LDS1 and LDS2 are the two most common subtypes of LDS and encompass mutations in TGFBR1 and TGFBR2, respectively. Less studied, yet still important, are mutations in SMAD and TGFB2, which lead to LDS3 and LDS4, respectively [49].

TGF-β signaling plays a critical role in blood vessel development, cell differentiation, proliferation, homeostasis, and vascular maintenance [50] and all four subtypes of LDS have been shown to affect TGF-β signaling. Significant volumes of research have investigated the molecular mechanisms regarding how alterations in TGF-β pathways lead to aneurysms [51]. Paradoxically, although many of the TGFBR1/2 mutations seen in LDS1/2 are predicted to reduce TGF-β activity, the opposite is seen. High levels of byproducts of TGF-β signaling, such as increased SMAD2 phosphorylation, have been observed in patients with LDS1/2 [52]. Some hypotheses have been suggested to explain this effect, including that a dysregulated negative feedback loop through a non-canonical TGF-β pathway or excess signaling through an intact TGFBR homodimer. However, more research is needed to definitively explain these fluctuations in TGF-β signaling as they relate to aneurysm development. See the following reviews for extensive investigation of this paradox [42,53,54].

Mutations in SMAD3, the intracellular effectors of the TGF-β pathway, have been causatively linked to LDS3. 67 pathogenic mutations of SMAD3 have been identified. In total, 61% of SMAD3 mutations linked to LDS are missense mutations, followed by frameshift mutations (23%) [54]. Upon phosphorylation by TGFBR1, SMAD3 transmits signals to the nucleus upon association with SMAD4. Similar to other forms of LDS, mutations in SMAD3 have been associated with increased TGF-β signaling, as demonstrated by increased levels of nuclear translocation of SMAD3 and connective tissue growth factor (CTGF) [47].

The clinical course of LDS patients tends to be more severe than patients with MFS. Patients with LDS1/2 have had significant vascular complications reported at a young age, some within the first year of life, significant risk of dissection below 5.0 cm aortic diameter, and high mortality with a median life expectancy of 37 years [46]. Guidelines for the management of LDS have thus recommended surgical intervention between 4.0 and 4.2 cm, although recent commentary has suggested that this may be too aggressive [15].

### 2.3. Ehlers-Danlos Syndrome (EDS)

Mutations in the collagen type III, alpha 1 gene (*COL3A1*), when associated with syndromic features such as facial dysmorphia, visible veins through translucent skin, and easy bruising, are common manifestations of EDS type IV, otherwise known as vascular EDS (vEDS) [55]. vEDS has been shown to be inherited in an autosomal dominant manner. It is 100% penetrant, and accounts for 5–10% of all EDS cases [56,57]. Over 700 unique mutations of *COL3A1* associated with vEDS have been identified, 50% of which are believed to be de novo mutations [58,59]. vEDS is rare, with an estimated prevalence of 1 in 150,000, but the high frequency of vascular complications means that morbidity and mortality are high [60]. Around 0.5–1% of patients develop unruptured aneurysms and an additional 4% develop intercranial hemorrhages [61,62]. Aortic dissection accounts for approximately 22% of all deaths in patients with vEDS, and these dissections were fatal 68% of the time [63].

The molecular mechanism of EDS is linked to its causative genetic mutation: alterations in the amount of functional collagen that is present. The main functional elements within type III collagen are [Gly-X-Y]_343_ repeats within triple helical regions; missense glycine substitutions in this region result in non-functional, dominant negative mutations [64]. Other mutations, such as splice-site variants, frame-shift, nonsense, and deletions have also been linked to EDS [63,65]. Mouse models of *COL3A1* mutant mice revealed nearly absent collagen in media of the aorta and approximately one-third the number collagen fibrils in the adventitia compared to wild type [66]. As a result of these mutations, collagen assembly in arteries and the GI system fails, limiting accommodation to mechanical stress, which increases rupture and dissection risk [67]. Gastrointestinal and other organ rupture is also comorbid due to the same pathophysiology [68].

The clinical course of patients with EDS is fraught with complications from unstable vasculature prone to rupture. The median survival for all patients with EDS is 51, with females living slightly longer than males [63]. The median age of first complication is 29 years old and 80% of patients with EDS will experience a complication (vascular or GI) prior to the age of 40 [57,69]. Genetics have shed some light on patients who tend to have better outcomes from EDS. Contrarily to previous belief, individuals with HI mutations may actually have more prevalent aortic complications compared to glycine substitutions or other splice site variants [59,65,70]. Patients with mutations in the N or C terminus of *COL3A1* or missense variants in the triple helix region may have milder cases of EDS but their vascular risk may remain high [69]. However, substantial heterogeneity between patients with identical mutations hampers clear genotype–phenotype associations that can be used in clinical practice [56]. Prognostication is further complicated by the fact that dissection can occur in patients with EDS can occur at midsized arterial diameters without aneurysm development, prompting a recommendation for prophylactic surgery at aortic diameters between 4.5 cm and 5.0 cm [71].

### 2.4. Other Causes of Syndromic TAAD

Several other genes have been linked to developing cases of syndromic TAAD, including *BGN* (Meester-Loeys syndrome) [72], *EFEMP2* (Cutis laxa, AR type Ib) [73,74], *ELN* (Cutis laxa, AD type) [75], and *SLC2A10* (Arterial tortuosity syndrome) [76] among others. While the molecular and cellular pathophysiologies of these conditions are complex and diverse, patients with these conditions do not consistently experience dissection or rupture at aortic diameters less than the recommended 5.0 cm aortic diameter for surgical intervention [71]. Therefore, from a clinical standpoint, these patients are not as concerning as the aforementioned syndromic causes and do not require accelerated, aggressive surgical management.

## 3. Causes of Non-Syndromic TAAD

Understanding the causes of non-syndromic TAAD has advanced significantly after decades of study. Because 21% of TAAD patients have another family member that is affected, positive family history significantly increases the severity of TAAD and the likelihood of aortic dissection (Figure 3). A genetic link for such cases was assumed and extensively investigated [18,77]. Importantly, this 21% estimate of heritability is likely an underestimate due to a lack of routine imaging that detects non-symptomatic aneurysms in family members [78]. Those without family members afflicted by TAAD—termed sporadic TAAD—were once thought to be entirely degenerative and not genetically mediated. However, the current research suggests that there may also be genetic mechanisms underlying sporadic TAAD due to variably penetrant gene expression [79]. Moreover, for many of these seemingly isolated cases, the proband may be the first in a subsequent line of inheritance. Other potential reasons why family history of TAAD might go unnoticed include asymptomatic aneurysms not detected or diagnosed, small families with few family members with which to trace inheritance over time, and a lack of sharing of health information amongst family members.

Familial TAAD appears to be more virulent than sporadic TAAD: age at first presentation is younger (58.2 vs. 65.7) and growth rates are faster (0.21 vs. 0.16 cm/year) [8] (Figure 3). Sporadic TAAD accounts for approximately 80% of all TAAD cases [80]. The ten-year mortality rate for non-syndromic TAAD is comparable to MFS (8.7% vs. 7.8%, respectively) [81]. Notably, genetic causality between syndromic and non-syndromic disease is difficult to separate; genes linked to syndromic TAAD (for example, *FBN1* as previously discussed) are also known to cause non-syndromic TAAD [82] without other systemic manifestations of the mutation. Several genes have been linked to familial non-syndromic TAAD. Four play a key role in contraction of the vascular smooth muscles: ACTA2, MYLK, MYH11, and PRKG1 [83,84]. Similarly to the causes of syndromic TAAD, the mutations in these four genes have been demonstrated to produce dissections at aortic diameters <5.0 cm [85].

Sporadic TAAD remains a challenge to detect, diagnose, and manage, as many cases do not have as clear a link to genetic causes as familial or syndromic TAAD. Approximately 10% of early age sporadic aortic dissection cases have a mutation in causative syndromic or familial TAAD genes [79]. In total, 28% of cases had one or more variants of unknown significance [79]. Lifestyle factors such a hypertension also seem to play a significant role in sporadic TAA [86].

Here, we review the aforementioned four genes that cause TAAD at aortic diameters <5.0 cm and briefly discuss other genes that have been linked to non-syndromic TAAD and sporadic TAAD.

### 3.1. ACTA2

*ACTA2* encodes SMC specific α-actin, which polymerizes to form thin filaments in the smooth muscle contraction element [87]. *ACTA2* is the most common mutation linked to familial TAAD, with about 50% penetrance and accounts for between 12% and 21% of all cases seen, by far the most common genetic defect linked to familial TAAD to date [83,84,88,89]. The cumulative risk of aortic dissection or aortic repair by 85 years old is 76%, approaching rates seen in cases of syndromic TAAD [90]. The median age of dissection is 35 years old, and more type A dissections are found compared to type B (54% vs. 24%) [90]. However, unlike other familial TAAD pathologies, penetrance does not seem to vary with age [88].

A majority of the *ACTA2* mutations that have been identified are DN mutations that prevent proper polymerization of actin or ATP hydrolysis that mediates SMC contraction [83]. Missense and premature truncating mutations have also been identified [83,91]. Aortic tissues in *ACTA2* mutant patients are characterized by proteoglycan accumulation, decreased and disorganized SMCs, and fragmented elastic fibers [83]. Increased insulin growth factor 1 (IGF-1) expression has also been noted, possibly as a compensatory mechanism from the instability that *ACTA2* mutations cause [92]. Patients also tend to have higher rates of occlusive vessel disease, leading to strokes and coronary artery disease, and one mutation has been linked to multisystemic smooth muscle dysfunction [89,93]. This evidence suggests a widespread and systemic disruption in adaptation to vascular wall stress, leading to aneurysm and dissection among other pathologies [94]. Surprisingly, mouse models have shown that simply deleting *ACTA2* does not lead to aneurysm [95]; aneurysms developed only upon exposure to angiotensin II, indicating an interplay between genetics and environmental factors in the development of TAAD [96].

Notably, *ACTA2* mutations tend to cause aortic dissection without preexisting aneurysm. As a result, one third of patients with *ACTA2* mutations have acute Type A dissections at aortic diameters <5.0 cm [97]. Pregnant women with *ACTA2* mutants are at even higher risk of small-diameter dissection, with dissections occurring between 3.8 and 4.7 cm in one study [97]. Patients with *ACTA2* mutations are therefore recommended for surgical intervention between 4.5 and 5.0 cm.

### 3.2. MYLK

Myosin light chain kinase (MLCK), encoded by *MYLK*, is another gene that has been causatively linked to non-syndromic TAAD. MLCK is known to phosphorylate regulatory light chain (RLC), which, in turn, increases myosin II ATPase activity, endocytosis, and stress fiber response [98]. Phosphorylation of RLC by MLCK also influences SMC contraction and allows for fine control of arterial pressure [99].

Two *MYLK* mutations were shown to produce a significant reduction in MLCK activity via a HI mechanism that decreased MLCK activity by 85%, resulting in impaired SMC contraction [98]. Further mechanistic studies in mice linked a 50% reduction of MLCK activity to a 40% reduction in RLC phosphorylation and associated contraction of the aorta [100]. Although most vessels are able to withstand a 40% reduction in contraction capacity, the aorta is exposed to the highest forces in the body, making it the first vessel to demonstrate a pathological phenotype [100]. In terms of genotype-phenotype correlations, missense mutations were linked to a higher risk of Type A dissection compared to null mutations [101]. Another family with a MYLK mutation showed differential phenotypes between heterozygous and homozygous patients [102]. *MYLK* mutations associated with TAAD have lower penetrance compared to other syndromic and non-syndromic mutations; only 38% of mutant individuals in one study experienced aortic events [101].

Due to a high risk of dissection at diameters <5.0 cm, the current guidelines suggest surgical intervention between 4.5 and 5.0 cm. Further prospective validation regarding this suggestion is needed.

### 3.3. MYH11

*MYH11* encodes smooth muscle myosin heavy chain (SM-MHC), a contractile component produced by SMCs [103]. Much like mutations in *ACTA2*, mutations in *MYH11* results in dysregulation of SMC contraction, which is critical for maintaining aortic wall stability.

*MYH11* mutations are thought to cause 1% of all familial TAAD cases [104]. This is inherited in an autosomal dominant fashion [9,105]. Specific mutations in the C-terminal coiled-coil region of SM-MHC have been linked to TAAD [103] and evidence suggests a DN mechanism [105]. The phenotype of these mutations—increased pulse wave velocity and decreased aortic compliance—demonstrates decreased elasticity in the aorta, which is linked to SMC dysfunction, and eventually leads to dissection [103]. Histological analysis of patients with *MYH11* mutations reveals focal fibromuscular dysplasia in the vaso vasorum, which may be suggestive of an inflammatory immune response consistent with what has been observed in other TAAs [104,106]. In addition, dysregulation of IGF-1 and angiotensin II signaling suggests a distinct mechanism from other TGF-β related mutations and may explain the increased incidence of occlusive vascular pathology observed in these patients; specifically, IGF-1 signaling has been linked to both increased production of contractile proteins in SMCs and SMC proliferation [104]. One study also demonstrated that *MYH11* mutations did not segregate with TAAD in one Dutch pedigree, suggesting that other unidentified genetic modifiers may play a role in the development of aneurysm [105]. Currently, surgical intervention is suggested at 4.5–5.0 cm [85,104].

### 3.4. PRKG1

*PRKG1* encodes the type I cGMP-dependent protein kinase (PKG-1), which controls SMC contraction by regulating phosphatases that dephosphorylate the RLC [87]. The major isoform of PKG-1 that is present in SMCs, PKG-1α, is activated by increased cGMP levels [107]. A gain-of-function mutation associated with the cGMP binding site (Arg177Gln) structurally changes PKG-1α to the point where its inhibitory domain is no longer active, leading to continuous activation [108]. This activation results in decreased RLC phosphorylation, relaxing SMCs [87]. Recently, a case report revealed another PRKG1 mutation associated with the ATP binding domain that was also associated with aortic dissection [109].

Clinical data on *PRKG1* mutations are limited but suggest a devastating aortic phenotype. In the one variant that has been identified to cause TAAD, penetrance of dissection is high (63%) and aortic enlargement was seen as good (37%). The average age at dissection was 37 years old, with the youngest occurring at 17 [108]. Similar to ACTA2, widespread vascular disease was noted in patients, including abdominal aneurysms, coronary aneurysms, coronary dissections, and tortuosity of the thoracic aorta [108]. Of the two patients with aortic diameters measured at the time dissection, one was noted at 4.3 cm [108]. Therefore, limited data notwithstanding, patients with PRKG1 are encouraged to have prophylactic surgery at 4.5–5.0 cm.

### 3.5. Other Genes Associated with Familial TAAD

Many other genes have been implicated in familial TAAD. These include the forkhead transcription factor (*FOXE3)* [110], lysyl oxidase (*LOX*) [111], methionine adenosyltransferase II α (*MAT2A*) [112], microfibril-associated glycoprotein 2 (*MFAP5*) [113], and notch homolog 1 (*NOTCH1*) [114]. Although patients with mutations in these genes tend to be at significantly higher risk for aortic aneurysm, they tend to follow the typical clinical course of aortic dilation followed by dissection at aortic diameters >5.0 cm. Therefore, standard guidelines regarding surgical intervention at aortic diameters between 5.0 and 5.5 cm are recommended to be coupled with genetic screening of family members.

Work over the past five years has identified additional genes that have implicated in causing familial TAAD. These include *ROBO4* (roundabout guidance receptor 4), a gene that contributes to endothelial function [115], ARIH1 (Ariadne RBR E3 Ubiquitin Protein Ligase 1), a gene involved in nuclear localization and morphology [116], and two genes in the TGF-β signaling cascade, LTBP1 (latent TGF-β binding protein 1) and LTBP3 (latent TGF-β binding protein 3) [117,118]. Due to the fact that these are recent discoveries, more investigation needs to be done to determine the clinical outcomes of these patients.

### 3.6. Genetics of Sporadic TAAD

Recent work investigating the genetic risk factors of sporadic TAAD has generated further insights into this seemingly random disease that accounts for a large majority of cases of non-syndromic TAAD [7]. A cohort of 765 patients with sporadic TAAD had a GWAS peak at *FBN1* even though patients did not present with any syndromic features of MFS, suggesting that even common variants in *FBN1* can place patients at a higher risk for TAAD [20]. Novel variants in low-density lipoprotein receptor-related protein 1 (*LRP1*), a protein involved in endocytosis and intracellular signaling, and unc-51-like kinase 4 (*ULK4*), involved in endocytosis and axon growth, were also identified in another large cohort study, further suggesting that other genes may play a role [119]. A recent study used blood gene signatures to identify and validate four genes—*CLU* (Clusterin), *DES* (Desmin), *MYH10*, and *FBLN5* (Fibulin 5)—related to sporadic TAA [120]. Another recent study found that 9.3% of sporadic TAAD patients had a pathogenic variant in genes associated with familial or syndromic TAAD (*TGFBR*, *COL3A1*, *SMAD*, *ACTA2*, *TGFB2*, or *TGFBR2*), indicating that genetic screening would benefit these patients and their families [79]. In addition, higher rates of variants of unknown significance (VUS) compared to control genomes suggests low yet stochastic penetrance and may be triggered by other physiologic factors such as hypertension [79]. Therefore, further research needs to be conducted on identifying non-syndromic mutations that place patients at higher risk for TAAD along with investigating screening strategies to detect these individuals [121].

## 4. Genetic Testing—Past and Future

Due to the virulence of TAAD, early detection, diagnosis, and intervention for high risk aneurysms would significantly decrease the morbidity and mortality that is observed upon dissection [122,123,124]. Therefore, genetic screening has been used as a reliable tool to identify high-risk patients early in order to ensure that they are closely monitored and receive timely prophylactic surgery if needed.

Genetic panels conducted via Sanger sequencing that directly sequence known causative genes for TAAD were once considered the standard of care to identify high risk patients [125]. However, the advent of economical next generation sequencing (NGS) techniques such as whole genome sequencing (WGS) has greatly amplified the ability of clinicians to find TAAD associated mutations [109,126,127,128]. Routine genetic screening via WGS is now being routinely done perioperatively to help guide surgical management as well as for families who have a history of TAAD. These mutations are then classified according to the likelihood that the variant is damaging. Current standards suggest classifying genes into five different categories—pathogenic, likely pathogenic, uncertain significance, likely benign, and benign—based off of experimental, computational, functional, and population data [129]. As the number of patients who are sequenced has gone up, the number of pathogenic causative mutations has steadily increased (Figure 4) [29,130].

Results from these studies show that between 71% and 87% of all patients screened did not have medically meaningful mutations, either likely benign or benign [29,131,132]. However, routine screening has resulted in the discovery of novel variants of unknown significance (VUS) that have unclear implications for the clinical management of patients [29]. Indeed, in the present era of whole exome sequencing (WES), most pathogenetic genes originate as VUS when first identified. Managing patients with VUS and to a certain extent, likely pathogenic variants (LPV), is difficult, as they exhibit variants in a gene associated with TAAD but the exact risk is unknown, either due to the lack of previously reported cases or ambiguous experimental evidence. Physicians’ opinions vary on whether or not to disclose or discuss VUS and similarly, patients have varying beliefs on whether they want to be informed about VUS [133].

Of particular note for VUS related to TAAD genes, recent evidence suggests that many variants that are classified as VUS may be pathogenic in a low-penetrant and less damaging fashion than pathogenic variants [121]. For example, two VUSs in the *LOX* gene showed a 16% decrease in enzymatic activity compared to wild type while known pathogenic variants showed up to a 50% decrease in activity. Similar results were seen in the *MYLK* gene. However, other VUS did not show a phenotype in animal models. The authors conclude that a new category—low-penetrant “risk variants”—may be appropriate to capture this kind of variable influence on patient phenotypes [121]. In summary, genetic screening helps increase the body of knowledge that allows for further understanding of clinically significant variants but currently has direct clinical utility for a minority of patients, although this may change with further research. Even when causal mutants are identified, the heterogeneity of pathologies from these genes hampers “one size fits all” clinical management. Moving beyond a monogenic view of TAAD to consider other multifactorial genetic or environmental mortifications of disease risk will also help with risk stratification [127].

Another potential genetic test to identify TAAD involves quantifying RNA signatures in the blood as a biomarker and screening tool. In a study with 94 individuals, RNA signatures were able to classify TAAD patients compared to controls with high sensitivity (72%) and specificity (90%) [134]. This approach is inexpensive, has a higher accuracy compared to other potential biomarkers, and can be utilized as a point-of-care diagnostic (Figure 5) [135]. Replication studies using this approach are presently underway.

miRNAs (MicroRNAs) have also been investigated as biomarkers for TAAD. In a recent study, an miRNA microarray assay identified miR-574-5p as a biomarker that was successfully able to differentiate between TAA patients and controls in a small cohort [136]. Another study used high-throughput miRNA sequencing on a small cohort of TAA patients and found two plasma miRNAs, miR-122-3p and miR-483-3p, that also showed sufficient discrimination for use as a biomarker [137]. Interestingly, the same study found striking sex-related differences in miRNA profiles with more dysfunction noted in males and noted that Krüppel-like factor 4 (KLF4), an important regulator of vascular SMCs, had significantly increased expression in TAA patients [137]. Both studies show how miRNA biomarkers, if validated in a clinical setting, could be a useful for detection of TAAD.

Determining the indications for routine genetic testing to detect thoracic aneurysms will need to consider the aforementioned limitations. One potential screening program is outlined in Figure 6 [138]. Testing lower risk patients and family members will lead to both false positives and false negatives due to the low penetrance and heterogeneity of many non-syndromic TAAD associated mutations. However, screening of family members of patients who have an early onset of TAAD (<50 years), family history of TAAD, and no history of hypertension may provide the highest yield. This group is suggested due to the high pretest probability of TAAD [124,139]. In addition, with the proliferation of direct to consumer genetic tests, more patients will be empowered with genetic information that could inform them of their risk for TAAD [140]. Clinicians will need to navigate this new world of genetic testing and appropriately counsel patients with variable penetrance and phenotypes of different TAAD associated mutations in mind.

When doing genetic testing, it is important to remember that several genes that predispose patients to higher risk for TAAD also places patients at risk for other arteriopathies, such as AAA, cerebral aneurysms, or coronary artery aneurysms. Some genes that have been specifically implicated in causing other arteriopathies outside of the thoracic aorta include FBN1, LOX, PRKG1, SMAD2, SMAD3, TGFB3, TGFBR1, and TGFBR2, although experimental evidence is weak for some genes [9,141,142,143]. Therefore, it is important to do a thorough evaluation of both the thoracic and abdominal aorta if a high-risk TAAD mutation has been identified, as well as other potentially affected arteries such as the cerebral arteries.

## 5. Conclusions

Dramatic strides have been made in understanding the underlying genetic risk factors for TAAD in the past three decades since the discovery of *FBN1* as the causative variant for MFS [27]. Since that time, causes of both syndromic and non-syndromic TAAD have been extensively investigated and described. Because intensive search for effective medical treatments for TAA has not been successful over decades [144], elucidating the underlying genetics of TAA offers the best prospects for future identification of truly effective, biologically based, medical therapies. In the future, linking patient-specific mutations to well defined phenotypes—a genomic dictionary linking mutations to outcomes—will enable precision management and finely tuned surgical intervention, limiting unnecessary surgeries while simultaneously decreasing mortality [9,145].

## Figures and Tables

**Figure 1 biomolecules-10-00182-f001:**
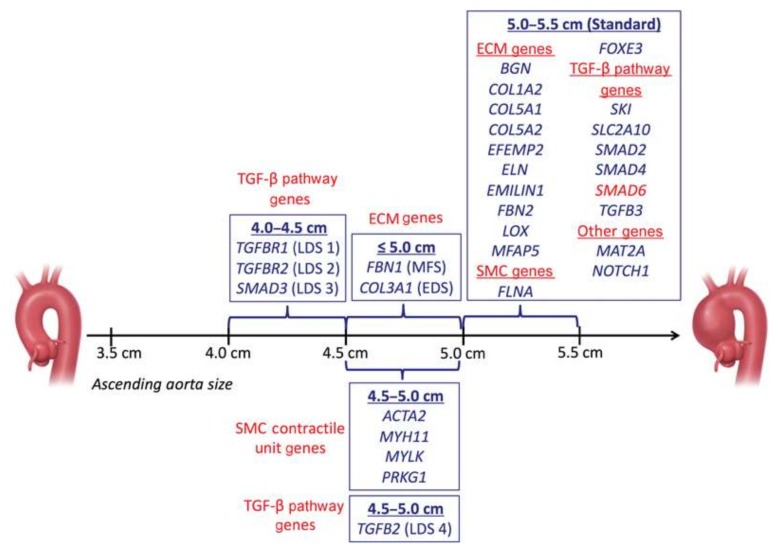
Ascending aortic dimensions for prophylactic surgical intervention. Abbreviations: ECM, extracellular matrix; SMC, smooth muscle cell; TAAD, thoracic aortic aneurysm and/or dissection; TGF, transforming growth factor. © 2020 by Thieme Medical Publishers Inc. [9]

**Figure 2 biomolecules-10-00182-f002:**
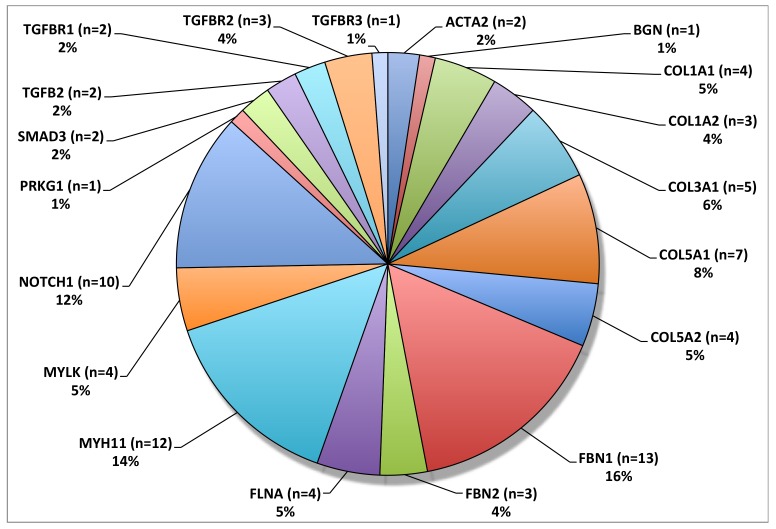
Overall frequency distribution of TAAD associated variants. FBN1 was the most frequently mutated gene in this patient cohort. © 2020 by Edi.Ermes. [30]

**Figure 3 biomolecules-10-00182-f003:**
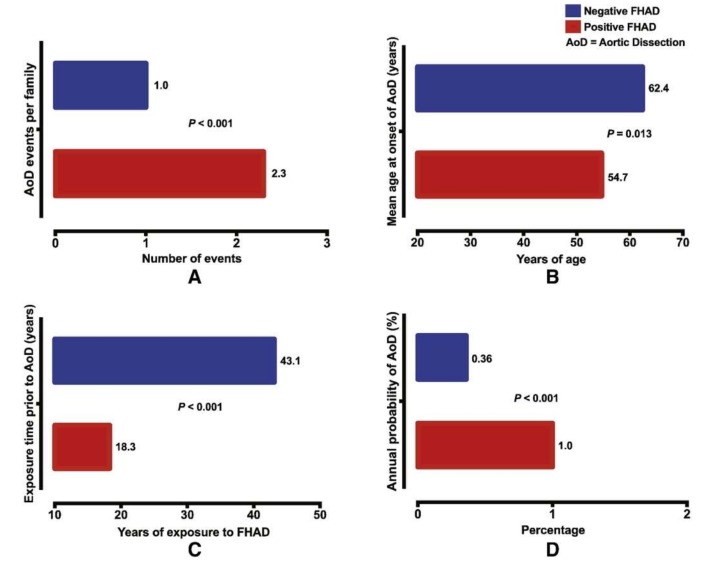
Positive family history of aortic dissection (FHAD) significantly increases the risk of developing a new dissection in unaffected family members, with (**A**) higher number of dissection events; (**B**) a younger age at dissection; (**C**) shorter duration of exposure prior to dissection; and (**D**) higher annual probability of aortic dissection. © 2020 by Elsevier. [77]

**Figure 4 biomolecules-10-00182-f004:**
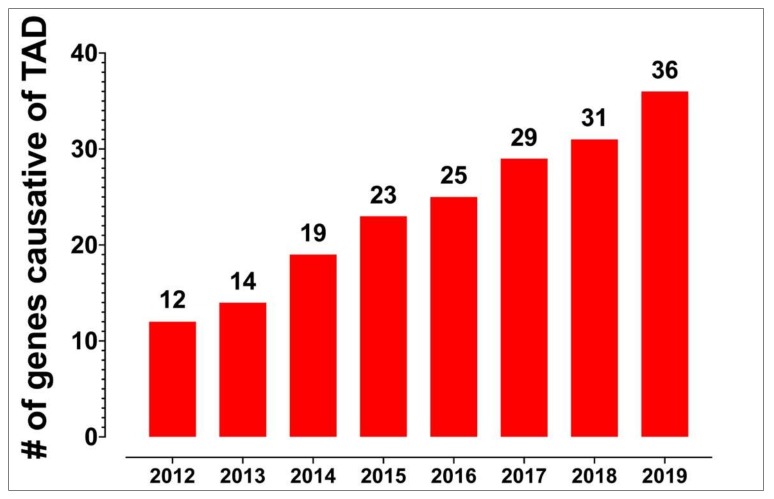
The number of mutations identified to be causative of TAAD has steadily increased since 2012. © 2020 by Thieme Medical Publishers Inc. [9]

**Figure 5 biomolecules-10-00182-f005:**
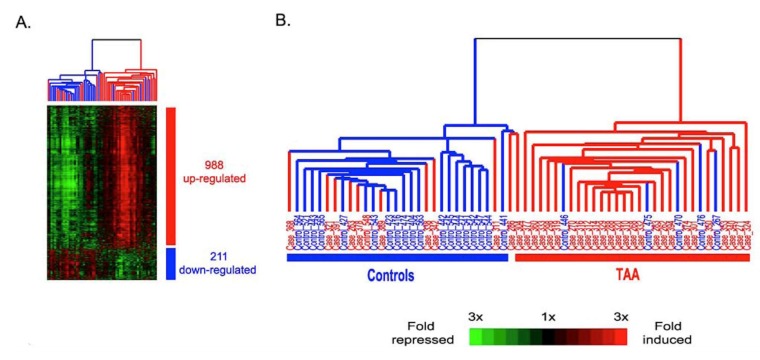
The data demonstrate that RNA signatures show promise as a biomarker for TAAD. A) Scaled down representation of the entire cluster of the 1199 signature genes and 61 whole blood samples. B) Experimental dendrogram displaying the clustering of the samples into two main branches: the TAA branch (red) and the control branch (blue) with only a few exceptions. © 2020 by Wang et al. [134]

**Figure 6 biomolecules-10-00182-f006:**
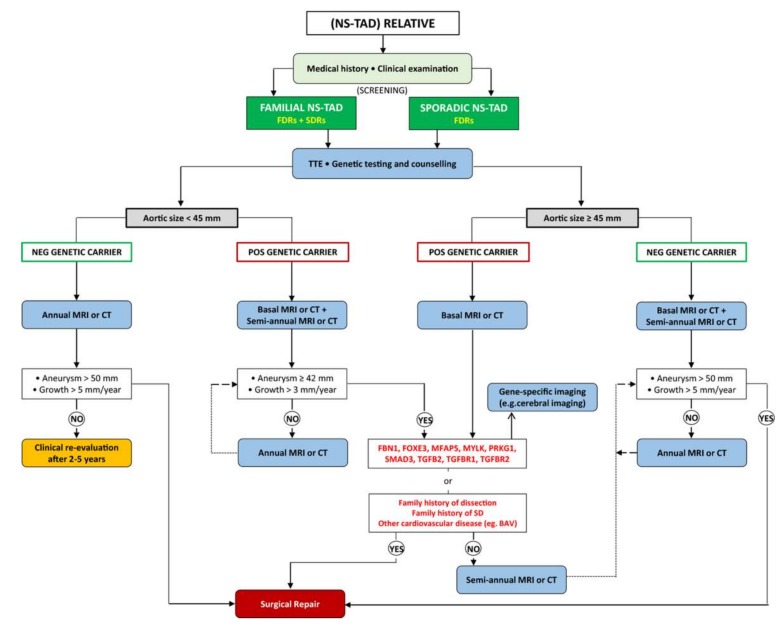
Proposed flow chart for a dedicated screening program for relatives of patients affected by non-syndromic diseases of the thoracic aorta. Note the importance of genetic screening in stratifying patients. BAV indicates bicuspid aortic valve; CT, computed tomography; FDRs, first-degree relatives; MRI, magnetic resonance imaging; NS-TAD, non-syndromic thoracic aortic disease; SDRs, second-degree relatives; TTE, transthoracic echocardiogram. © 2020 by Mariscalo et al. [138]

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
