# Peer review of "The Genetics of Thoracic Aortic Aneurysms and Dissection: A Clinical Perspective"

_biomolecules, 2020, doi:10.3390/biom10020182_

Round 1
Reviewer 1 Report
The review is timely, accurate, rather complete and well written. Now that attention for the genetic causes for aortic aneurysms in general is growing, a quality overview like this one, is very welcome, in particular given the increasing possibilities to find a genetic cause with the introduction of WES and WGS sequencing.
A small comment regarding acronym for marfan syndrome, tradiosnally MFS, in the manuscript MFN or MFS. please adjust.
The flowchart seems very important for the clinical use. However, it is very unfortunate that it is not readible, unless enlarged. Here a solution is needed to make fig 6 readable, and the information better accesible for the audience, especially when printing the manuscript in a normal way.
The message that genetic aneurysm cannot be excluded in cases without a family history of aneurysm is of great importance.
Therefor , to put more emphsis on this subject, I would suggest that the authors elaborate a bit more on the causest, so that the audiance understand better the reasons for familial disease to go unnoticed. For instance explain that familial aneurysm may be missed because, asymptomatic aneurysm in relatives may go undiagnosed and relatives will not know they have an aneurysm. Also when families are small, chance of having affected relatives are smaller. And it is possible that within families information on disease is not shared.
Another issue that is not addressed and should be included is the current molecualr classification of variants in aneurysm genes, and how to deal with likley pathogenic ( (L)PV) variants , pathogenic (PV) variants and variants of unkown clinical significance (VUS)
Because in daily practice much more VUS than (L)PV are reported and we have to figure out how to deal with this kind of results, for patients and their families.
In line with this , authors need to indicate how the variants in the diagram in fig 2 are classified.
Mutation in COl3A1 present mostly with midsized arterail ruptures, not with TAAD, that should be included.
Finally, but very important, is to explain that all the gens involved in TAAD may also present with isolated abdominal aneurys in relatives. For some gene, abdominal aneurysm occur even more frequently that in the thoracic part (ie TGFB3) and reports show contribution of mutations in AAA patients.
Among the major benefits of finding a genetic cause, apart for genotype phenotype correlations that may help to predict outcome and guide clinical management , is that high risk for aneurysm in family members can accurately be predicted by presymptomatic testing of relatives.
if a genetic cause may present with thoracic as well as abdominal aneurysm, this had major implication for family screening and the relatives should be advised to have full aortic imaging (CT or MRI) instead of only the thoracic part of the aorta.
I would therefor emphasis that the authors incude the occurence of abdominal aneurysm and extens the recommedations accordingly in this otherwise thorough overview so that ppossible life threatening abdominal aneurysm may not go unnotice in the TAAD families.
Author Response
A small comment regarding acronym for Marfan syndrome, traditionally MFS, in the manuscript MFN or MFS. please adjust.”: We have updated the abbreviation from MFN to MFS in all locations in the draft. “The flowchart seems very important for the clinical use. However, it is very unfortunate that it is not readable, unless enlarged. Here a solution is needed to make fig 6 readable, and the information better accessible for the audience, especially when printing the manuscript in a normal way.”: We have enlarged the figure to the margins of the page, which should make the figure legible for the reader. “The message that genetic aneurysm cannot be excluded in cases without a family history of aneurysm is of great importance. Therefore , to put more emphasis on this subject, I would suggest that the authors elaborate a bit more on the causest, so that the audience understand better the reasons for familial disease to go unnoticed. For instance explain that familial aneurysm may be missed because, asymptomatic aneurysm in relatives may go undiagnosed and relatives will not know they have an aneurysm. Also when families are small, chance of having affected relatives are smaller. And it is possible that within families information on disease is not shared.”: We thank the reviewer for this useful and insightful comment. We have added these thoughts to the first paragraph of section 3 (lines 352- 355). “Another issue that is not addressed and should be included is the current molecular classification of variants in aneurysm genes, and how to deal with likely pathogenic ( (L)PV) variants , pathogenic (PV) variants and variants of unknown clinical significance (VUS). Because in daily practice much more VUS than (L)PV are reported and we have to figure out how to deal with this kind of results, for patients and their families.”: We thank the reviewer for pointing out this important clinical aspect of aortic aneurysms that we did not include in our original draft. We added a description of how mutations are classified with a detailed reference [130] and included a more in depth discussion of how to manage patients who have VUS or LP variants (line 553-559) and expanded upon some intriguing new research about variable phenotypes of VUS in TAAD (lines 576 - 585).
“In line with this, authors need to indicate how the variants in the diagram in figure 2 are classified.”: While we recognize that knowing the distribution of pathogenic, likely pathogenic, VUS, etc. for each gene would be useful information but unfortunately the data from which this figure is derived was not granular enough to capture this information and therefore we are unable to include classifications in this figure. “Mutation in COL3A1 present mostly with midsized arterial ruptures, not with TAAD, that should be included”: We clarified this point in line 301. “Finally, but very important, is to explain that all the gens involved in TAAD may also present with isolated abdominal aneurysm in relatives. For some gene, abdominal aneurysm occur even more frequently that in the thoracic part (ie TGFB3) and reports show contribution of mutations in AAA patients. Among the major benefits of finding a genetic cause, apart for genotype phenotype correlations that may help to predict outcome and guide clinical management , is that high risk for aneurysm in family members can accurately be predicted by presymptomatic testing of relatives. If a genetic cause may present with thoracic as well as abdominal aneurysm, this had major implication for family screening and the relatives should be advised to have full aortic imaging (CT or MRI) instead of only the thoracic part of the aorta. I would therefor emphasis that the authors include the occurrence of abdominal aneurysm and extend the recommendations accordingly in this otherwise thorough overview so that possible life threatening abdominal aneurysm may not go unnoticed in the TAAD families.”: We thank the author for pointing this out and we agree that patients who have genes involved in TAAD should be examined for other arteriopathies, including in the abdominal aorta. We have included a discussion of this in lines 637 to 644.
Reviewer 2 Report
The major disappointment about this review is low presentation levels of novelty. The authors mainly discuss scientific data which is known for almost decades. In addition the reference list almost doesnt include the newest data in the field and needs to be updated (2017-2019). Furthermore, the content of the review is mainly based on clinical data and the molecular mechanisms of the disease remain poorly described.
Therefore, due poor representation of novelty, I strongly suggest to reject it.
Author Response
The authors mainly discuss scientific data which is known for almost decades. In addition the reference list almost doesn’t include the newest data in the field and needs to be updated (2017-2019).”: We do recognize that some recent research was not included this manuscript. In response to the reviewer’s recommendation, we have added several new studies that were not included in the first draft, including Arnaud et al. 2017 [30], Seo et al. 2018 [34], Tan et al. 2017 [39], Boileau et al. 2019 [138], Chen et al. 2019 [40], Shalata et al. 2018 [104], Gould et al. 2019 [117], Tan et al. 2018 [118], Quiñones-Pérez et al. 2018 [119], Guo et al. 2018 [120], Xu et al. 2019 [122], Arnaud et al. 2019 [134], and Lawal et al. 2018 [135]. “Furthermore, the content of the review is mainly based on clinical data and the molecular mechanisms of the disease remain poorly described.”: We expanded upon some mechanistic studies that underly the development of TAAD in this paper, including a discussion of the role of TGF-β in MFS (lines 177 - 184), some mechanistic evidence for vEDS (lines 284-286), and additional theories about the development of TAAD in patients with MYH11 mutations (lines 458 - 460).
Reviewer 3 Report
The review by Ostberg N.P. et al is very well written and formulated. The review summarizes the current data on the genetics of thoracic aortic aneurysms very well, especially the clinical aspects, which is the aim of the authors. Particularly noteworthy here is the good definition and differentiation of syndromic, non-syndromic and sporadic forms of aneurysms. Genetic mutations which are causative for the development of aneurysms are clearly presented and their effects are associated with a possible clinical effect (mortality, aneurysm growth, age of onset).
Overall, this review is very good, but I have a few small comments:
I find the title somewhat misleading in comparison to the nomenclature used in the rest of the manuscript, because only the aneurysms are mentioned there and then often both in the manuscript - perhaps the dissections should also be mentioned in the title. Some small spelling mistakes have to be corrected, e.g. Line 43: "are diagnosed"; Line 87: Abbreviation for smooth muscle cells = SMC; Line 134: no distance between "diameter" and "in" Line 156: "regarding" Conclusion: I find this paragraph very important, perhaps it is also worth mentioning that the results of the genetic studies could not only have an effect on the surgical rehabilitation, but also on the drug treatment of the patients in order to be able to treat a progression of the dilatation better.
Author Response
“I find the title somewhat misleading in comparison to the nomenclature used in the rest of the manuscript, because only the aneurysms are mentioned there and then often both in the manuscript - perhaps the dissections should also be mentioned in the title.”: The reviewer’s point is well taken. We have altered the title to be consistent with the nomenclature used throughout the rest of the manuscript. “Some small spelling mistakes have to be corrected, e.g. Line 43: "are diagnosed"; Line 87: Abbreviation for smooth muscle cells = SMC; Line 134: no distance between "diameter" and "in" Line 156: "regarding””: We have corrected the noted spelling errors. Thank you. “Conclusion: I find this paragraph very important, perhaps it is also worth mentioning that the results of the genetic studies could not only have an effect on the surgical rehabilitation, but also on the drug treatment of the patients in order to be able to treat a progression of the dilatation better.”: In response to the reviewer’s astute observation, we have added the following sentence to the conclusion (lines 649 to 651): “Because intensive search for effective medical treatments for TAA has not been successful over decades, elucidating the underlying genetics of TAA offers the best prospects for future identification of truly effective, biologically based, medical therapies.”
Reviewer 4 Report
The authors submitted a review of the current knowledge of the genetics of thoracic aneurysms and dissection (TAAD). The review is non-exhaustive but highly informative as it describes and refer to key original studies identifying mutations linked to TAAD. A few additions and modifications are suggested to improve clarity and flow of the review:
1) The statement made line 102-103, in this context in particular, could be confusing to certain readers. It could be interpreted as follows: Patients with FBN1 mutations (i.e.: MFS patients) make up the most of TAAD patients. However, not all patients with FBN1 mutations are diagnosed with MFS. In fact, several studies (including studies published by the authors) have shown that mutations in FBN1 are most common in sporadic TAAD, as the authors mention later in the review.
2) The section on LDS contains many typos, stylistic and grammatical errors and could use a few extra sentences to ensure smoother transitions between ideas. For example: the 4 LDS subgroups are mentioned but not defined while later in the section they’re referred to as LDS1/2, LDS3, etc.
3) The authors tend to use TAA and TAAD interchangeably in the introduction in particular. In some instances, specifically when they refer to original research articles, it is important to make the distinction between aneurysms and dissection. For example, line 55: TAA would fit better than TAAD. Same goes for many other instances, please review the manuscript for accurate use of TAAD vs TAA. Another example can be found in the first paragraph of section 3 where family history of TAAD is described and illustrated by figure 3, which refers to data pertaining to dissection only, and not aneurysms.
4) Line 162: the explanation as to why the loss of function mutation in TGFBR leads to an excessive activation of the receptor must be more developed to be understood by the readers. As it is stated, the reader cannot understand that homodimers of TGFBR1 and 2 assemble to form an heterotetramer and in the case of LDS1/2, the loss of function mutation of TGFBR1 leads to homotetramerization of TGFBR2 and its excessive activation (and vice versa). Please expand this point to make it clear to the reader.
5) Line 182-183: what are the common manifestations of vEDS mentioned by the authors in this sentence?
6) Line 185: With regard to the 700 mutations identified, were these identified in all EDS patients? Or only in vEDS patients? Similarly, the numbers described in this paragraph: do they refer to all EDS patients? Or vEDS patients specifically? If they refer to all EDS patients, it would be informative to describe similar demographics for vEDS patients.
7) Please refrain from citing other reviews articles to support data reporting or interpretation. Cite original research article instead.
8) Line 284: it should be noted that MLCK phosphorylation of RLC is also key in SMC contraction and that’s how MLCK allows for fine control of blood pressure.
9) Line 293-295: This statement is highly misleading. The SMCs in the aorta do not govern the aortic diameter as they do in muscular or resistance arteries. The aortic caliber is mainly dependent on the biomechanical and material properties of the aorta (expansion/recoil and elastin/collagen content, respectively). Plus, the way the aorta counteracts the highest forces in the body is through the load-bearing adventitial layer enriched in collagen, not the SMC contraction. Please revise or remove this statement.
Minor:
Line 12: revise sentence (“a solely a degenerative disease”)
Line 15: TAAD are syndromic and non-syndromic, not genes – revise the sentence.
Line 32: replace “remains” with “remain”
Line 57: remove “causes of” to read “systemic TAA account for approximately…”
Line 60: modify the beginning of the sentence to “Cases of non-syndromic TAA...”
Line 61-62: should read “can be explained by mutations in genes involved in syndromic TAA”.
Line 63: the preposition “termed familial TAA” seems out of place
Line 76: provide a citation to support this statement.
Line 83: change to “Causes of syndromic TAAD”
Line 86: change “diseases” to “dysfunction”.
Line 93: should read “Marfan Syndrome (MFS)” (and change all the subsequent MFN to MFS). Similarly, refrain from using “Marfans”, please refer to Marfan patients instead.
Line 100: add a reference to support the statement that FBN1 mutation was linked to MFS in 1991.
Line 124: replace “liked” by “linked”
Line 129: Figure 1 does not illustrate this statement.
Line 182: though --> through
Line 212 and 221: TAAD are syndromic/non-syndromic, not the causes. The titles should read “Other causes of syndromic TAAD” and “Causes of non-syndromic TAAD”. Please check other instances throughout the manuscript where syndromic/non-syndromic were used as an attributive adjective to nouns other than TAAD.
Legend figure 3: define FHAD
Line 230: please revise “low penetrant gene expression”, it does not seem like an accurate way to describe low penetrance of gene mutations in sporadic TAAD (if that’s what the authors mean in this sentence)
Line 246: ACT2 --> ACTA2
Line 275-276 “dilated aneurysm” is a redundancy.
Line 281: patient’s --> patients
Line 269: define IGF-1
Line 350: define and briefly describe LRP1 and ULK4
Line 404: outline --> outlined
Author Response
“The statement made line 102-103, in this context in particular, could be confusing to certain readers. It could be interpreted as follows: Patients with FBN1 mutations (i.e.: MFS patients) make up the most of TAAD patients. However, not all patients with FBN1 mutations are diagnosed with MFS. In fact, several studies (including studies published by the authors) have shown that mutations in FBN1 are most common in sporadic TAAD, as the authors mention later in the review.”: We recognize that this statement was confusing, especially to an audience not familiar with the nuances of syndromic and non-syndromic TAAD. We added a sentence clarifying that that FBN1 mutations cause both syndromic and non-syndromic TAAD. We also clarified that FBN1 was the most mutated gene in a specific whole exome sequencing study of both syndromic and non-syndromic patients. “The section on LDS contains many typos, stylistic and grammatical errors and could use a few extra sentences to ensure smoother transitions between ideas. For example: the 4 LDS subgroups are mentioned but not defined while later in the section they’re referred to as LDS1/2, LDS3, etc.”: We added a description that clarified what subtypes of LDS corresponded with each gene mutation. Rather than exhaustively discussing the reasons behind the complicated paradoxical TGF-β signaling, we referred readers to three reviews that covered the material in an effort to keep the scope of the review focused. We can add more material to this section if the reviewer so suggests. “The authors tend to use TAA and TAAD interchangeably in the introduction in particular. In some instances, specifically when they refer to original research articles, it is important to make the distinction between aneurysms and dissection. For example, line 55: TAA would fit better than TAAD. Same goes for many other instances, please review the manuscript for accurate use of TAAD vs TAA. Another example can be found in the first paragraph of section 3 where family history of TAAD is described and illustrated by figure 3, which refers to data pertaining to dissection only, and not aneurysms. ”: We now recognize that we were not as careful as we should have been regarding using the abbreviations TAA and TAAD. We updated line 55 as requested. While Figure 3 does reference dissection, the inclusion criteria for this study were patients with thoracic aortic disease, meaning they all had preexisting aortic aneurysm prior to dissection. Therefore, we believe that TAAD is most appropriate in this case. Several other changes between TAAD, TAA, and aortic dissection were made in the manuscript, particularly in the Introduction. “Line 162: the explanation as to why the loss of function mutation in TGFBR leads to an excessive activation of the receptor must be more developed to be understood by the readers. As it is stated, the reader cannot understand that homodimers of TGFBR1 and 2 assemble to form an heterotetramer and in the case of LDS1/2, the loss of function mutation of TGFBR1 leads to homotetramerization of TGFBR2 and its excessive activation (and vice versa). Please expand this point to make it clear to the reader.” As mentioned above, rather than exhaustively discuss the reasons behind paradoxical TGF-β signaling, we referred readers to three reviews that covered the material in an effort to keep the scope of the review focused. We understand the reviewer’s point. We feel such detailed explanations as the reviewer’s expertise allows may be excessive for this review manuscript. We can add more information if the reviewer so suggests. “Line 182-183: what are the common manifestations of vEDS mentioned by the authors in this sentence? “: The common manifestations of vEDS are facial dysmorphia, visible veins through translucent skin, and easy bruising, which were referred to in the previous sentence. This sentence was edited for clarity. “Line 185: With regard to the 700 mutations identified, were these identified in all EDS patients? Or only in vEDS patients? Similarly, the numbers described in this paragraph: do they refer to all EDS patients? Or vEDS patients specifically? If they refer to all EDS patients, it would be informative to describe similar demographics for vEDS patients. “: These number are referring to patients specifically with vEDS. We have edited for clarity. “Please refrain from citing other reviews articles to support data reporting or interpretation. Cite original research article instead. “: The following references were changed to original investigations rather than reviews: [7], [13], [14], [30] (updated with link to mutation database instead), [34], [39], [63], and [103]. Thank you. “Line 284: it should be noted that MLCK phosphorylation of RLC is also key in SMC contraction and that’s how MLCK allows for fine control of blood pressure. “: We thank the reviewer for this insight and we have edited our draft to make this point more clear. “Line 293-295: This statement is highly misleading. The SMCs in the aorta do not govern the aortic diameter as they do in muscular or resistance arteries. The aortic caliber is mainly dependent on the biomechanical and material properties of the aorta (expansion/recoil and elastin/collagen content, respectively). Plus, the way the aorta counteracts the highest forces in the body is through the load-bearing adventitial layer enriched in collagen, not the SMC contraction. Please revise or remove this statement.”: We thank the reviewer for pointing out this inaccuracy in the manuscript. The misleading statement has been removed from the manuscript. “Minor revisions…”: All changes were incorporated as suggested. Thank you
Round 2
Reviewer 2 Report
The authors have vigorously improved the manuscript. Therefore, i dont see no more obstacles for further evaluation of the manuscript. However:
minor suggestion for the authors for the final point:
The authors discuss quantifying RNA signatures in the
blood as a biomarker and screening tool and present some examples (miR-574 as one). In addition, the authors introduce tgf-beta signaling cascade as a core signaling pathway during TAAD development. Therefore, with regard to the novelty and further improvement of the present manuscript, the authors should cite newest in the field Gasiule et al, 2019 (https://doi.org/10.3390/jcm8101609) as they identified miRNA expression profiles in TAA tissue and blood plasma samples using high-troughput miR-seq, demonstrated that changes of miRNA expression were not associated directly between tissue and plasma samples miRNA expression alterations in ascending thoracic aortic aneurysm tissues, showed that miRNAs in TAA tissues are differently expressed in sex dependent manner (more significantly in male patients), validated new biomarkers, and related the TAA miRNAs with TGF beta signaling cascade components (at target gene and protein expression levels) and potential master regulator of TAA pathology - KLF4.
Please, introduce this in to the manusript.
Author Response
Dear Biomolecules Editorial Board and Reviewers,
We would like to thank the reviewers and editors once again for their helpful feedback on our review article. In response to Reviewer #2’s comments, we have made the following changes/modifications:
“The authors discuss quantifying RNA signatures in the blood as a biomarker and screening tool and present some examples (miR-574 as one). In addition, the authors introduce tgf-beta signaling cascade as a core signaling pathway during TAAD development. Therefore, with regard to the novelty and further improvement of the present manuscript, the authors should cite newest in the field Gasiule et al, 2019 (https://doi.org/10.3390/jcm8101609) as they identified miRNA expression profiles in TAA tissue and blood plasma samples using high-throughput miR-seq, demonstrated that changes of miRNA expression were not associated directly between tissue and plasma samples miRNA expression alterations in ascending thoracic aortic aneurysm tissues, showed that miRNAs in TAA tissues are differently expressed in sex dependent manner (more significantly in male patients), validated new biomarkers, and related the TAA miRNAs with TGF beta signaling cascade components (at target gene and protein expression levels) and potential master regulator of TAA pathology - KLF4. Please, introduce this in to the manuscript.”: We thank the reviewer for pointing out this recent publication regarding miRNAs. We have included this study in our manuscript in lines 621-627.
Thank you,